# Metabolic Adaptation-Mediated Cancer Survival and Progression in Oxidative Stress

**DOI:** 10.3390/antiox11071324

**Published:** 2022-07-05

**Authors:** Yongquan Tang, Zhe Zhang, Yan Chen, Siyuan Qin, Li Zhou, Wei Gao, Zhisen Shen

**Affiliations:** 1Department of Pediatric Surgery, West China Hospital, Sichuan University, Chengdu 610041, China; yqtang@scu.edu.cn; 2State Key Laboratory of Biotherapy and Cancer Center, West China Hospital, West China School of Basic Medical Sciences & Forensic Medicine, Sichuan University, Collaborative Innovation Center for Biotherapy, Chengdu 610041, China; scuzz@stu.scu.edu.cn (Z.Z.); yanchen0524@scu.edu.cn (Y.C.); 2017324060025@stu.scu.edu.cn (S.Q.); 2015224060079@stu.scu.edu.cn (L.Z.); 3Clinical Medical College & Affiliated Hospital of Chengdu University, Chengdu University, Chengdu 610106, China; 4Department of Otorhinolaryngology and Head and Neck Surgery, The Affiliated Lihuili Hospital, Ningbo University, Ningbo 315040, China

**Keywords:** metabolic reprogramming, oxidative stress, metabolic adaptation, cancer stemness, tumor metastasis

## Abstract

Undue elevation of ROS levels commonly occurs during cancer evolution as a result of various antitumor therapeutics and/or endogenous immune response. Overwhelming ROS levels induced cancer cell death through the dysregulation of ROS-sensitive glycolytic enzymes, leading to the catastrophic depression of glycolysis and oxidative phosphorylation (OXPHOS), which are critical for cancer survival and progression. However, cancer cells also adapt to such catastrophic oxidative and metabolic stresses by metabolic reprograming, resulting in cancer residuality, progression, and relapse. This adaptation is highly dependent on NADPH and GSH syntheses for ROS scavenging and the upregulation of lipolysis and glutaminolysis, which fuel tricarboxylic acid cycle-coupled OXPHOS and biosynthesis. The underlying mechanism remains poorly understood, thus presenting a promising field with opportunities to manipulate metabolic adaptations for cancer prevention and therapy. In this review, we provide a summary of the mechanisms of metabolic regulation in the adaptation of cancer cells to oxidative stress and the current understanding of its regulatory role in cancer survival and progression.

## 1. Introduction

Redox homeostasis is essential to maintain the normal structure and functions of cellular components, but oxidative stress frequently occurs in cancer cells as a result of oncogene activation, hypoxia, inflammation, and therapeutics [1,2,3,4]. Abrupt accumulation of reactive oxygen species (ROS) has detrimental effects on various components of cancer cells, leading to cellular dysfunction or even cell death [5,6,7,8]. In particular, metabolic enzymes are sensitive to ROS, with the most noted examples being glyceraldehyde-phosphate dehydrogenase (GAPDH) and pyruvate kinase M2 (PKM2) in the glycolytic pathway [9]. The wide application of 2-flourine-18[(18)F] fluoro-2-deoxy-D-glucose (FDG) positron emission tomography–computed tomography (PET–CT) demonstrated the glycolytic phenotype in most cancers. Therefore, ROS-induced oxidation and inactivation of GAPDH and PKM2 can cause the depression of both aerobic and anaerobic glycolysis, leading to decreased proliferation and/or cell death due to shortages of energy and tricarboxylic acid cycle (TCA)-derived biosynthesis, especially in cancer cells in early stages that are more dependent on glycolysis [10,11,12]. Despite these reports linking overt ROS damage to metabolic pathways and other cellular components, it is noteworthy that cancer cells also adapt to such overwhelming ROS levels and metabolic impairment [13,14]. It has been well documented that the oxidative pentose phosphate pathway (PPP) and the synthesis of reduced glutathione (GSH) are enhanced, largely contributing to the production of nicotinamide adenine dinucleotide phosphate (NADPH) and GSH, the most prominent antioxidant molecules [15,16]. On the other hand, cancer cells tend to recruit carbon flux from lipids and glutamine into nucleotide synthesis through the non-oxidative PPP [17,18,19] and into TCA-coupled oxidative phosphorylation (OXPHOS) and biosynthesis [20,21], which meet substrate and energy requirements. This relies on the wide crosslinks in the metabolic pathways of glucose, lipids, and amino acids. This metabolic regulation is known to be driven by a complex network consisting of several metabolic modulators, including nuclear factor erythroid 2-related factor 2 (NRF2), hypoxia-inducible factors (HIFs), forkhead box proteins (FOXOs), nuclear factor kappa-B (NF-κB), and/or RAC-alpha serine/threonine-protein kinases (AKTs). Their activations depend on ROS levels and their specific inducers, suggesting the heterogeneity of metabolic adaptation under different pro-oxidant conditions.

Accordingly, metabolic regulation plays a central role in cancer adaptation to oxidative stress. Mounting evidence has shown that metabolic regulation, involving the activation of different metabolic modulators with oncogenic properties, metabolic reprogramming, and optimized ROS levels, is tightly linked with cell fate decisions in cancer [22,23,24,25]. A better understanding of how cancer cells orchestrate these metabolic modulators to achieve stress adaptation has potential implications for developing redox- and metabolism-targeting therapeutic strategies.

## 2. Oxidative Stress in Cancer Cells

Oxidative stress in cancer cells is induced by various endogenous or exogenous pro-oxidant elements, such as hypoxia, inflammation, and numerous therapeutics. Upon these stimuli, overwhelming ROS may be produced through a host of oxidoreductases in several compartments of cells, primarily including mitochondria, cytoplasm, and the endoplasmic reticulum (ER), inter alia [6,26].

Mitochondria contribute the most to both physiologically and pathologically endogenous ROS [9,27,28]. Numerous studies have revealed that cancer cells exhibit remarkable plasticity in metabolic phenotypes [29,30], with the most notable example being glucose catabolism, which impacts endogenous ROS generation [31,32]. The selective switch to anaerobic glycolysis in rapidly proliferating cancer cells, even under sufficient nutrition and O_2_ (Warburg effect), favors the acquisition of a ‘pro-oxidant state’ partly due to the diversion of pyruvate away from the mitochondria [33,34,35]. Despite the previously held notion that cancer cells selectively rely on the Warburg effect, there is ample evidence to indicate that cancer cells can switch between glycolysis and OXPHOS to cater to impending energy demands [33,34,36]. To this end, high OXPHOS-coupled aerobic glycolysis is also a potential source of increased ROS levels in cancer cells [37,38,39]. For example, CEM leukemia and HeLa cervical cancer cells overexpressing BCL-2 were shown to have increased OXPHOS and mitochondrial ROS generation [40,41,42]. This suggests that both hypo-functional and/or hyper-functional mitochondria are linked to the increased generation of ROS. Intriguingly, various antitumor drugs, as well as radiation, induce significant oxidative distress in cancer cells [43,44,45,46], due in part to the impairment of mitochondrial function and metabolism [40,45,47] (Figure 1). For example, the doxorubicin, bleomycin, or platinum coordination complexe causes mitochondrial DNA damage or prevent DNA synthesis by inducing cellular oxidative distress [47,48]. Furthermore, genetically unstable clones generated upon irradiation also displayed higher intracellular ROS levels, potentially due to reduced mitochondrial activity and respiration [49,50]. In addition, cancer cells under hypoxia are associated with an increase in ROS, likely because of the deficiency in O_2_ that prevents electron transfer across the mitochondrial complexes, thereby increasing the possibility of electron leakage to generate ROS [27,51,52]. These findings indicate that the plasticity of mitochondrial function, under the control of various pro-oxidant elements, contributes significantly to ROS regulation in cancer cells.

Cytoplasmic ROS-producing enzymes are represented by members of the NADPH oxidase (NOX) family, such as NOX1-4, whereby cancer cells respond to various extra-cellular stimuli [53,54]. Several studies have shown the role of ROS generation in cancer cells, particularly from the standpoints of oncogene activation, chemoresistance, survival, inflammation, and metastasis. For instance, NOXs can be activated by a wide variety of ligands, such as tumor necrosis factor (TNF), angiotensin II, platelet-derived growth factor (PDGF), and pro-epidermal growth factor (EGF) [55,56] (Figure 1). Furthermore, the function of NOX1 in generating superoxide anion radicals (O_2_^•−^) was shown to be critical in RAS-mediated cell transformation [57]. Similarly, the introduction of the dominant negative mutant (N17) of GTPase RAC1, a subunit of NOX, reduced superoxide levels, thus inhibiting the growth of mutant KRAS-driven cells [58].

The ER is also a ROS factory where nascent proteins are folded and modified to become functional [59]. Disulfide bond formation is essential for the primary structure of proteins and is catalyzed by protein disulfide isomerase (PDI). In a normal state, the oxidized PDI can be reduced by ER oxidoreductases, represented by oxidoreductase 1 (ERO1), to generate H_2_O_2_ as a byproduct [60], which maintains a high basic level of ROS in ER [61,62]. Furthermore, ROS production and release from the ER are significantly increased during ER stress [63,64]. When misfolded proteins accumulate beyond a tolerable threshold within the ER, an unfolded protein response (UPR) is induced to mobilize protein-folding capacity and otherwise to promote cell apoptosis [60,65]. ER stress is frequently documented in chemotherapies and radiotherapies because DNA and enzyme damage basically produce a large amount of abnormal proteins [66,67]. In addition, NOX4 and NOX5 were found in the ER and act, in a way, as their cytoplasmic members. However, the restoration of ROS levels with the endogenous antioxidant N-acetylcysteine (NAC) did not significantly prevent cancer cell death [68], indicating that ER stress-derived ROS seldom causes cancer cell death directly.

## 3. Metabolic Stress Due to Overwhelming ROS

Excessively high ROS levels can induce damage to many cellular macromolecules, including DNA, lipids, and proteins, which are essential for the cellular functions of cancer cells [69,70]. Numerous metabolic enzymes are especially sensitive to oxidative stress, the impairment of which results in metabolic stress.

Glycolysis and its coupled TCA and mitochondrial OXPHOS are essential for the rapid proliferation of cancer cells, especially during tumorigenesis or macro-metastasis [10,71]. However, several glycolytic enzymes are vulnerable to elevated ROS levels and can be inactivated by redox modification. For example, GAPDH, a key enzyme in glycolysis that catalyzes the first step of the pathway by converting glyceraldehyde 3-phosphate (GA-3P) into 3-phospho-glyceroyl phosphate (3P-G), can be redox-modified and -inhibited to form sulfenic acids or S-glutathionylated at Cys152 in response to oxidative stress [72], which leads to decreased carbon flux into anaerobic and aerobic glycolysis, as well as endogenous serine and one-carbon synthesis. Similarly, pyruvate kinase M2 isoform (PKM2), a glycolytic limiting enzyme catalyzing phosphoenolpyruvate (PEP) to form pyruvate, can undergo oxidative modifications and inhibition at Cys358, thus leading to the depression of carbon flux into lactate synthesis and TCA [62,73]. The inhibition of GAPDH and PKM2 collaboratively causes a deficiency of carbon source-supported ATP generation, lactate secretion, and TCA-driven biosynthesis (Figure 1). In addition, the activity of the mitochondrial trifunctional protein subunit β (TPβ), a critical enzyme in fatty acid oxidation that catalyzes 3-ketoacyl-CoA to form acetyl-CoA, can be inhibited by Cys458 oxidation, thus restraining the carbon compensation to TCA from fatty acids [74,75]. Accordingly, oxidative stress can impair glycolysis and TCA function, thereby leading to proliferation arrest or even death of cancer cells.

## 4. Metabolic Adaptation in Response to Oxidative and Metabolic Stresses 

ROS detoxification is carried out by a group of non-enzymatic molecules and enzymes represented by GSH, NADPH, thioredoxin (Trx), and superoxide dismutase (SOD). Among these, SODs are antioxidant enzymes that catalytically dismutate O_2_^•−^ into H_2_O_2_ [76,77]. The Trx/thioredoxin reductase (TrxR) system exerts a similar mechanism to GSH in ROS scavenging [78,79], mediating repair of oxidized proteins [80]. The antioxidative capacity of these enzymes is determined by their expression, which is transcriptionally activated by certain transcriptional factors, indicating that they may not be involved with timely response to oxidative distress. In contrast, the GSH-correlated system is the most prominent and rapidly regulated antioxidant defense mechanism, comprising the antioxidative molecule GSH that reduces H_2_O_2_ and ROOH into H_2_O and ROH, respectively. Weighted equally with GSH in this antioxidant system is NADPH, a molecule mainly derived from the oxidative PPP [81], which reduces oxidized glutathione (GSSG) back to GSH through the action of glutathione reductase (GR) [9,82]. In addition to ROS scavenging, cancer cells are likely to strengthen carbon flux from lipids and/or glutamine into TCA and lactate, thus meeting essential energy and biosynthesis demands. Notably, these adaptive metabolic changes are performed, alternatively, through the activation or inactivation of metabolic enzymes in certain steps that are systemically controlled by several metabolic modulators which also act as ROS sensors [83]. The activation of these metabolic modulators and their processes of metabolic remodeling are the mechanisms of adaptation to oxidative stress and metabolic impairment that are tightly linked with oncogene activation, therapeutic resistance, hypoxic resistance, and immune escape.

### 4.1. Activation of ROS Sensors and Metabolic Modulators

Compared with intricate metabolic reprogramming, upstream metabolic modulators, represented by NRF2, HIF, FOXO, NF-κB, and AKT, may present more druggable targets for modulation. ROS sensors and metabolic modulators can be activated by direct oxidative modification or through other ROS signaling, which processes depend on their specific inducers. For example, HIF pathways are generally induced by hypoxic stress and elevated ROS [84]. Similarly, NF-κB is mainly responsible for metabolic and redox responses to immune and inflammatory stimuli. On the other hand, the basal activity or expression of these ROS sensors and metabolic modulators has been wildly reported as being upregulated in cancers, which suggests their oncogenic properties that render an intrinsic capacity for metabolic adaptation in oxidative distress. These ROS sensors and metabolic modulators therefore play a central role in redox regulation by sensing ROS levels as well as driving gene expression to cause metabolic reprogramming, thus optimizing cancer cell survival and proliferation upon oxidative stress.

#### 4.1.1. NRF2

NRF2 is a non-specific ROS sensor and powerful metabolic modulator [85,86,87], which functions via a specific DNA sequence, referred to as the antioxidant response element (ARE), a cis-regulatory element found in the promoter region of several antioxidant genes, including SODs, as well as a number of critical enzymes and transporters in substrate and energy metabolic pathways [71,88]. NRF2 is known to be upregulated by ROS stimulation and oncogenic activation. The NRF2-Kelch-like ECH-associated protein 1 (KEAP1) system represents the paradigmatic ROS sensor apparatus [24,89]. Under resting conditions, NRF2 is constitutively degraded by the KEAP1-Cullin 3 (CUL3) E3 ligase complex. During oxidative stress, the KEAP1 system is directly modified by ROS, thereby leading to conformational changes in the KEAP1–CUL3 complex that disable the ubiquitination of NRF2, allowing nascent NRF2 to freely translocate into nuclei (Figure 2) [90]. In addition, there is mounting evidence showing the oncogenic activation of NRF2 in various malignant processes as a result of gene mutation and/or activation of other oncogenes or signaling pathways governing the transcription or degradation of NRF2, thereby conferring on cancer cells a more reduced intracellular environment [91,92]. Interestingly, DNA sequencing of human-derived and lab-induced tumor tissues revealed that both NRF2 and KEAP1 are commonly mutated in the region encoding their binding domain, resulting in dissociation and subsequent degradation of NRF2 [93,94,95,96]. Additionally, some malignant markers, such as DPP3, PAQR4, and BRCA1, were found to competitively bind to KEAP1 or NRF2, thus interfering with their interaction (Figure 2) [97,98,99]. Notably, the transcription of NRF2 can be activated by several oncogenes, such as KRAS, B-RAF, and c-MYC, to elevate its basal level, mediating oncogene-induced tumor initiation and progression [99,100,101]. A recent study also showed that the removal of a post-translational modification, glycation, from NRF2, mediated by Fructosamine-3-kinase, is critical for the oncogenic function of NRF2 [102]. NRF2 expression and activity in response to oxidative stress largely determine the metabolic program and antioxidant capacity of cancer cells [71,87,92]. A host of metabolic enzymes are transcriptionally upregulated by NRF2 (Figure 3), including enzymes in the oxidative and non-oxidative PPPs [71,103], serine/glycine biosynthesis [87], GSH synthesis, glutaminolysis [104], and lipolysis [105,106]. This suggests that NRF2 activation has multiple purposes in supporting energetic, anabolic, and antioxidant needs in response to oxidative stress.

#### 4.1.2. Hypoxia and HIFs

HIFs are a family of transcription factors that are activated in hypoxic states and stabilized by hypoxia-induced ROS [107,108,109]. They consist of three α-subunits (HIF1α, HIF2α, and HIF3α) and one β-subunit (HIF1β), which serves as a heterodimer with one α-subunit and the β-subunit [110]. During normoxia, the proline residues of HIFα are hydroxylated by prolyl hydroxylases (PHDs) in the presence of O_2_, iron (Fe^2+^), and α-ketoglutarate. Under hydroxylation, HIFα is polyubiquitylated by the von Hippel–Lindau tumor suppressor protein (pVHL) complex and guided for proteasomal degradation. Hypoxia prevents HIFα hydroxylation and pVHL-dependent degradation, allowing HIFα to translocate to the nucleus and dimerize with HIF1β to form an active transcriptional complex that binds to the hypoxia-response element (HRE) (Figure 2). Through this process, mitochondria-derived ROS likely facilitate the stabilization of HIFα, as genetically or pharmacologically decreasing ROS levels were shown to reduce HIFα protein level [51,111,112]. Although the mechanism underlying ROS-induced HIFα stabilization remains largely unknown, Fe^2+^ oxidization may be a reason [110]. 

Therefore, HIFs were not regarded as a strict ROS sensor and modulator until it was found that H_2_O_2_ treatment of HeLa cells enhanced the transactivation of HIF1 by stabilizing and re-distributing Sentrin/SUMO-specific protease 3 (SENP3) from the nucleoli to the nucleoplasm, thus promoting the SUMOylating and activity of p300, a co-activator of HIF1α [113,114]. In another study, NOX-mediated ROS production was shown to trigger HIF1α activation via the ERK pathway in breast cancer cells [115]. Furthermore, several subsequent studies on fibroblasts reported a positive effect of ROS on HIF1α translation [116,117]. Taken together, elevated ROS levels increase HIF activity via multiple processes, thereby regulating metabolic adaptation to hypoxia-linked oxidative stress as well as hypoxic stress. There is mounting evidence of metabolic modulations by HIFs, including increased uptake of glucose, fatty acids, and glutamine [18,20,118], and upregulated glycolysis, the non-oxidative PPP [18], lipid storage [118,119], GSH synthesis, and glutaminolysis [20] (Figure 3). Additionally, HIF-induced metabolic modulation is also mediated by NRF2, since the activation of HIFα was shown to promote nuclear localization of NRF2 [120,121]. Accordingly, HIFs are responsible for metabolic adaptation during oxidative stress under hypoxic conditions via supplying energetic, anabolic, and antioxidant demands.

#### 4.1.3. Inflammation and NF-κB

NF-κB was known to serve as a molecular lynchpin that links chronic inflammation to increased cancer risk [122]. Classical NF-κB signaling is activated by several IKK-activating cytokines, including TNF and IL-1, produced by macrophages and other immune cells, resulting in the crosstalk between cancer cells and the immune system [122,123]. Upon stimulation, NF-κB signaling is activated following the phosphorylation and degradation of the inhibitor of NF-κB (IκB) mediated by a IκB kinase (IKK) heterotrimer (IKKγ/IKKα/IKKβ). This allows NF-κB to translocate to the nucleus and activate gene transcription (Figure 2). Notably, an increase in ROS generation is invariably associated with immune activation and inflammation, which affects the activity of NF-κB [124,125]. However, ROS-induced oxidative modifications of different effectors may lead to either a positive or a negative effect on NF-κB activity [126,127]. An intricate crosstalk between NF-κB activation and intracellular ROS amplification has been implicated in various processes of carcinogenesis and/or cancer cell survival [128,129]. For example, NF-κB could potentially be involved in ROS production as it has been shown to control the expression of COX2 and 5-LPO, two sources of ROS generation [45]. In turn, NR-κB can be activated as a result of Zinc-doped copper oxide nanocomposite (nZn-CuO NP)-induced ROS production in the growth of human cancer cells [130]. Under H_2_O_2_ treatment, a homodimer of IKKγ can be formed through ROS-induced disulfide bonds, thus potentiating its downstream effect on NF-κB [131]. On the other hand, oxidation of IKKβ on Cys179 or oxidation of p52 on Cys62 may inhibit NF-κB [126]. Nevertheless, due to its ability to positively influence the transcription of antioxidant proteins, such as NRF2 and SOD, NF-κB is a redox-modulating transcription factor and this antioxidant property is believed to be, at least in part, responsible for its survival-promoting effect in cancer cells. As a metabolic modulator, NF-κB was found to cause metabolic reprogramming of anaerobic glycolysis by upregulation of GLUT3 upon deletion of p53, a transcriptional factor favoring mitochondrial respiration [132]. In a separate study, NF-κB knockdown was found to stimulate OXPHOS through the upregulation of mitochondrial synthesis of cytochrome c oxidase 2 (COX2), which was mediated by p53 activation [133,134]. As such, the function of NF-κB in metabolic remodeling has a tight link with p53. Furthermore, EGFR-induced activation of NF-κB was shown to interact with HIF1α, thus leading to the upregulation of glycolysis through transactivation of PKM2 [135]. Accordingly, NF-κB contributes to metabolic reprogramming to anaerobic glycolysis under oxidative stress and inflammatory stress.

#### 4.1.4. Nutrients and AKT

The PI3K-AKT pathway is one of the most commonly activated pathways in human cancers. It is activated in response to nutrient-correlated factors, such as insulin, growth factors, and various cytokines, thus regulating key metabolic pathways, including glucose metabolism and maintenance of redox homeostasis [17]. As ROS accumulation may stimulate AKT activation through phosphorylation and inhibition of tensin homolog (PTEN) [136,137], the PI3K-AKT pathway also acts as a ROS sensor and metabolic modulator during oxidative stress. AKT activates enzymes involved in the non-oxidative PPP and TCA-derived fatty acid synthesis, which indicates its role as a biosynthesis driver [138,139] (Figure 3). Notably, AKT also controls other metabolic modulators, including NRF2, HIFα, NF-κB, and FOXO [17] (Figure 2). In this way, FOXO was found to be inhibited due to phosphorylation by AKT, which may provide a feedback loop among ROS, AKT, and FOXO, ensuring a balance of FOXO activation. In addition, NRF2, HIFα, and IKKγ can be dephosphorylated and activated via the phosphorylation and inhibition of glycogen synthase kinase 3 (GSK3) by AKT [17]. Therefore, the PI3K-AKT pathway also plays a positive role in mediating metabolic regulation in oxidative stress.

#### 4.1.5. ASK1/JNK and FOXOs

Several studies have consistently revealed that FOXOs are required for normal stem cells to resist oxidative stress [22,140,141]. Long known as an effector in the PI3K/AKT pathway, FOXOs are involved in many cellular processes, including cell cycle regulation, apoptosis, and metabolism [142]. The human FOXO family consists of four members, FOXO1, FOXO3, FOXO4, and FOXO6, which are activated by ROS signaling but inhibited by the canonical insulin receptor through the PI3K/AKT pathway in the presence of growth factors [17,142]. Under oxidative stress, redox modifications of transportin-1 and FOXO4 result in interprotein disulfide-bridge formation and binding, which are required for nuclear localization and activation of FOXO4 [143] (Figure 2). Additionally, ASK1/JNK pathways mediate FOXO activation upon oxidative stress, in which the ROS-induced homodimer of ASK1 directly leads to the activation of JNK, a kinase predominantly facilitating the nuclear translocation and activation of FOXO through phosphorylation [144]. As a metabolic modulator, FOXO3 was shown to drive the transcription of groups of metabolic enzymes involved in glycolysis, the PPP, lipolysis, glutaminolysis, and GSH synthesis in normal stem cells (Figure 3), which was confirmed by FOXO3 knockout [145,146], though the role of FOXO as a metabolic modulator in oxidative stress has not been well-defined. Nevertheless, FOXOs have also been well defined as negative effectors of cancer stem cell (CSC) properties in various cancers [147,148,149,150,151], suggesting their tight relationship with metabolic modulation in cancer cells. It has been reported that FOXOs response to hypoxia-induced oxidative stress in cancer cells through suppressing nuclear-encoded mitochondria-regulating genes [152,153], which suggests a role for FOXO3 in cooperation with HIFs in serving metabolic adaptation to hypoxic oxidative stress. The effects of FOXOs on cellular ROS levels are debatable, with one study having showed that FOXO3 knockdown led to decreased ROS levels in cholangiocarcinoma cells, which also protected cells from oxidative-stress-induced cell death [154]. In this study, FOXO3 was found to promote the transcription of KEAP1, and this FOXO3–KEAP1 axis was downregulated with a corresponding hyperactivation of NFR2 in clinical cholangiocarcinoma samples [154]. In contrast, several other studies have demonstrated that ROS levels were elevated following FOXO3 knockdown in normal stem cells as a result of downregulation of genes involved in the OXPHOS, PPP, and GSH synthesis pathways [145,146]. These contradictory results indicate the potential functional differences of FOXOs in cancer cells compared with normal stem cells, which is likely dependent on their crosslinks with other ROS sensors and metabolic modulators. Nevertheless, exogenous H_2_O_2_ treatment was found to activate FOXO3 [155], while a separate study showed that FOXO3 depletion sensitized cholangiocarcinoma cells to H_2_O_2_-induced damage [154], demonstrating its role in controlling adaptation to oxidative stress.

### 4.2. Adaptive Metabolic Reprogramming

Metabolic adaptation during oxidative stress often depends on the collaboration of several metabolic modulators because more than one metabolic modulator will be activated even in a certain specific inducer. It has been well documented that the activation of these metabolic modulators by ROS can often be independent of their inducers [20,104,118,154]. For example, hypoxic oxidative stress may result in the activation of HIFs, NRF2, and FOXO [20,152,156], and p53 deletion-induced oxidative stress may give rise to NRF2 and NF-κB [132,157]. Furthermore, oxidative stress can be simply induced by H_2_O_2_ treatment in vitro, but it always occurs along with other stresses in vivo; in other words, oxidative stress is generally not the only consequence of its inducer, especially in hypoxia, which leads to metabolic impairment through oxidative stress and oxygen deficiency [158]. Nevertheless, these processes of metabolic remodeling also share common features, such as the potentiation of anaerobic glycolysis, the PPP, GSH synthesis, and TCA-linked biosynthesis, along with the inhibition of carbon flux from pyruvate to acetyl-CoA and OXPHOS.

#### 4.2.1. PPP Upregulation

The PPP contributes to most NADPH and pentose, providing essential antioxidants, as well as materials for the de novo synthesis of nucleotides, which is vital for cell proliferation and the repair of DNA damage in cancer cells during oxidative stress [159,160,161]. It was shown that drug or hypoxia resistance in CSCs correlated with elevated PPP activity and NADPH generation [34,149,162]. During oxidative stress, glucose uptake is increased as a result of the overexpression of glucose transporters (GLUTs) driven by FOXO, HIF1, NF-κB, and AKT [132,145,163]. Furthermore, the carbon flux of glucose in the course of glycolysis is powerfully directed into the PPP through the activation of several enzymes in both the oxidative and non-oxidative PPPs. For example, glucose-6-phosphate dehydrogenase (G6PD) and 6-phosphogluconate dehydrogenase (6PGD), the two rate-limiting enzymes of the oxidative PPP, can be transcriptionally upregulated by NRF2 and FOXO [71,145,164], thereby increasing the generation of NADPH to support ROS clearance and de novo nucleotide synthesis.

In addition, transaldolase (TALDO) and transketolase (TKT) in the non-oxidative PPP are also transcriptionally controlled by NRF2, HIF, and FOXO, leading to increased intracellular ribulose-5-phosphate (R-5P) [18,71,145]. In this process, TKT was also found to be phosphorylated and activated by AKT [139]. Meanwhile, the expression of phosphoribosyl pyrophosphate amidotransferase (PPAT) and methylenetetrahydrofolate dehydrogenase 2 (MTHFD2), two enzymes involved in the conversion of R-5P to inosine monophosphate (IMP), are also upregulated by FOXO and NRF2, consecutively [145,165]. The potentiation of the non-oxidative PPP provides abundant materials for the de novo synthesis of nucleotides.

#### 4.2.2. Reinforcement of Anaerobic Glycolysis

Reinforcement of anaerobic glycolysis is another common response of cancer cells to oxidative stress (Figure 3). Cancer cells tend to produce ATP through anaerobic glycolysis in response to oxidative stress, especially during hypoxia, potentially due to the fact that mtOXPHOS-linked ATP generation would produce additional ROS. With the increased uptake of glucose, as mentioned above, the expression of GAPDH and PKM2 can be activated by HIF1 and NF-κB [135,158,166,167], thus promoting the synthesis of glucose-derived pyruvate. Recently, it was found that the transcriptions of hexokinase 2 (HK2), an enzyme phosphorylating glucose to produce glucose-6-phosphate (G-6P), and GAPDH can be significantly upregulated through NRF2-dependent BTB and CNC homology 1 (BACH1) activation [168]. Meanwhile, ROS-caused GAPDH and PKM2 inactivation is also attenuated as a result of ROS scavenging. Furthermore, the increased pyruvate is more likely to be directed into lactate synthesis (Warburg effect) rather than TCA because pyruvate dehydrogenase (PDH), an enzyme catalyzing the synthesis of pyruvate-derived acetyl-CoA, is inhibited by PDH kinase 4 (PDK4), which can be phosphorylated and activated by AKT or transcriptionally activated by HIF1 and FOXO3 [20,145,152]. The conversion from pyruvate to lactate is further supported by the activation of lactate dehydrogenase A (LDHA), an enzyme that can be transcriptionally controlled by HIF1 [18,119]. In addition, separate studies have found that the increased pyruvate is also derived from malate, a substrate in TCA, due to the catalyzation of malic enzyme 1 (ME1), whose transcription is driven by NRF2 during oxidative stress [71,169]. Therefore, the reinforcement of anaerobic glycolysis during oxidative stress results in the increased lactate synthesis that is required for cancer growth [170,171]. This is consistent with the upregulated synthesis of one-carbon units, another group of materials essential for cell proliferation. This is mediated by the activation of several enzymes, such as phosphoserine aminotransferase 1 (PSAT1) [172], in the course of de novo synthesis of serine, indirectly driven by NRF2, which subsequently supplies the synthesis of one-carbon units under the catalyzation of serine hydroxymethyltransferase 2 (SHMT2) [87]. Therefore, the reinforcement of anaerobic glycolysis is vital to meet the energetic and anabolic demands of cancer cells during oxidative stress.

#### 4.2.3. Increased Utilization of Glutamine

Mounting evidence shows metabolic remodeling to depend more on glutamine in cancer under oxidative stress [173,174], one key reason being the increased GSH synthesis that is critical for cancer cell survival against overwhelming ROS, and another reason being the glutaminolysis which is a major source for TCA-derived biosynthesis. Several metabolic modulators were shown to be responsible for the activated GSH synthesis pathway, including NRF2, HIFs, and FOXO [20,71,145] (Figure 3). For example, the SLC1A5 and SLC7A11 (xCT) membrane transporters mediating the uptake of glutamine and cysteine, respectively, are positively regulated by NRF2 [157,174]. Furthermore, the glutamyl-cysteinyl ligase (GCL), the rate-limiting enzyme catalyzing GSH synthesis, is also transcriptionally activated by NRF2 [71,87,104]. In addition, the expression of SLC1A5 on both the cytomembrane and the mitomembrane, as well as the GLS1, an enzyme mediating the transformation of glutamine into glutamate, can be increased by HIFα [20,175,176].

On the other hand, separate studies showed potentiated glutaminolysis as a result of the activation of glutamate dehydrogenase (GDH), regulated by NRF2, HIF1, and/or FOXO during oxidative stress [104,145,177]. This leads to the production of α-ketoglutarate (α-KG), a central substrate fueling TCA-linked biosynthesis, including the syntheses of lactate, lipids, and pyrimidines, which are required for the proliferation of cancer cells due to the blockage of the native TCA source, pyruvate-derived acetyl-CoA, as discussed before (Figure 3). In particular, malate is increasingly converted into pyruvate given the transactivation of ME1 by NRF2, as mentioned above, which further facilitates lactate synthesis [71,169]. In addition, glutaminolysis-dependent syntheses of aspartate and its linked pyrimidine in B-cell lymphoma cells are upregulated as a result of the transactivation of aspartate aminotransferase 2 (GOT2) by NF-κB, thus supporting cell proliferation [178]. Interestingly, a recent study showed that GDH1-derived α-KG can directly bind to and activate IKKb and NF-κB signaling, thus promoting glucose uptake by upregulating GLUT1 [163]. Despite the reinforcement of TCA through glutaminolysis, OXPHOS is still suppressed due to the inhibited expression of ECT genes by FOXO3 and NF-κB, such as COX1 and COX2 [133,152,179,180]. Given this, metabolic remodeling to increase the utilization of glutamine is essential to meet ROS scavenging and anabolic demands in response to oxidative stress, thus favoring the survival and growth of cancer cells.

#### 4.2.4. Remodeling of Fatty Acid Metabolism

Fatty acids serve as another important carbon source fueling the TCA through β-oxidation, which can also be impaired by ROS, thereby resulting in ATP reduction [110,181]. The metabolic remodeling of fatty acids tightly depends on the oxygen state. In normoxia, ROS-impaired fatty acid oxidation can be reinforced by the activation of NRF2 and FOXO. For instance, the expression of CD36, a transporter importing fatty acids across mitochondrial membranes, is positively regulated by NRF2, contributing to increased fatty acid uptake into mitochondria [182]. Additionally, FOXO is found to promote the transcription of acetyl-CoA synthetase (ACSS), a critical enzyme mediating the synthesis of fatty acid-derived acetyl-CoA [145]. On the other hand, the de novo synthesis of fatty acids derived from citrate is inhibited as a result of attenuated expression of several intrinsic enzymes regulated by NRF2, such as ATP-citrate lyase (ACL), acetyl-CoA carboxylase 1 (ACC1), and fatty acid synthase (FAS) [164,183,184] (Figure 3).

Increased uptake and de novo synthesis of fatty acids results in lipid storage [118,185,186]. In particular, the expressions of both medium- and long-chain acyl-CoA dehydrogenases (MCAD and LCAD), enzymes initiating β-oxidation, are inhibited by HIF1 [119] (Figure 3). Another study showed that the glutamine-dependent synthesis of fatty acids is potentiated in cancer cells in response to hypoxia, which is mediated by HIF1-driven activation of E3 ubiquitin-protein ligase SIAH2, and leads to ubiquitination and degradation of α-KG dehydrogenase (OGDH) and blockage of the conversion of α-KG into succinate [185]. Instead, the glutamine-derived α-KG is reversely converted into citrate, which initiates the de novo synthesis of fatty acids [107,158]. In addition, the transcription of fatty acid binding proteins (FABPs) 3 and 7, two fatty acid transporters on the plasma membrane, are driven by HIF1, thus promoting fatty acid uptake from the exterior [118]. In the same study, the intracellular lipid storage of cancer cells is known to protect against ROS and support survival in hypoxia–reoxygenation [118].

## 5. Metabolic Adaptation-Mediated Cancer Progression

Redox regulation has long been associated with cancer initiation and progression [187,188]. The activation of ROS sensors and metabolic modulators is involved in the regulation of tumorigenesis or metastasis by metabolism-dependent and -independent ways.

### 5.1. Cancer Cell Stemness

ROS sensors and metabolic regulators were shown to have wide crosslinks with CSC regulators, such as SOX2 and CD44, as well as CSC-related signaling pathways, such as the Notch pathway and the Wnt pathway, thus contributing to the self-renewal and proliferation of cancer cells. In a *Kras*^G12D^-driven pancreatic ductal adenocarcinoma (PDAC) mouse model, upregulated NRF2 led to the MDM2-mediated activation of the Notch pathway, a signaling pathway involved in multiple aspects of CSC biology [189], thereby accelerating neoplastic progression [190]. Several HIF family members were validated to regulate the expression of SOX2 and MYC [107]. In turn, MYC expression was shown in another study to cause HIF-1α accumulation by enhancing OXPHOS and ROS generation [191], which suggests that HIFs stimulate the expression of CSC markers in a positive-feedback manner. Mounting evidence has validated that NF-κB mediates CSC induction and maintenance [192,193]. Further, NF-κB p65 was found to form transcriptional super-enhancers (SEs) at several CSC genes, including tumor protein p63 (*TP63*), MET Proto-Oncogene (*MET*), and FOS Like 1 (*FOSL1*) [194]. Additionally, NF-κB was reported to facilitate CSC properties by inducing the expression of lipid desaturases [195] and the secretion of multiple cytokines [196], resulting in enhanced tumor formation. In contrast, FOXOs have long been viewed as negative effectors of CSCs via downregulating β-catenin [155,197], SOX2 [198,199,200], and CD44 [151].

In addition, several ROS-induing metabolic properties play roles in CSC regulation. It was shown that NRF2-induced increased utilization of glutamine is required for the tumorigenesis of *Kras*-driven spontaneous lung adenocarcinomas in mice [174]. The upregulation of the PPP, another core event that promotes tumorigenesis, was shown to be associated with the activation of NRF2 and PI3K-AKT in non-small-cell lung cancer (NSCLC) mouse models [201]. Consistently, another study revealed that knockout of TIGAR, an enzyme supporting PPP activation, by hydrolyzing both F-2,6BP and F-1,6BP back into F-6P, reduced proliferation and intraepithelial neoplasia precursor lesions in *Kras*-driven PDAC mouse models [202]. Due to the demands of redox homeostasis and anabolic programs, PPP-fueled tumorigenesis is believed to rely on the intrinsic synthesis of NADPH and nucleotides [71,169,203]. Most notably, glucose and glutamine are known as major energy and material resources of cancer cells and it has been well documented that they play critical roles in CSC regulation, though this remains controversial. Numerous studies have demonstrated that quiescent CSCs are dependent on OXPHOS, which is generally fueled by aerobic glycolysis [34,191,204,205]. On the other hand, signaling that promotes metabolic dependency on anaerobic glycolysis was reported to reinitiate tumor formation, which is known to be driven by active CSCs (Figure 4A,B). For example, in response to p53 deletion, activation of anaerobic glycolysis was found to facilitate oncogene-induced cell transformation of mouse embryonic fibroblasts through a positive feedback loop with NF-κB signaling [132]. In contrast, HK2 deficiency-induced suppression of anaerobic glycolysis markedly inhibited tumor development and extended lifespan in a mouse model [138]. In this context, anaerobic glycolysis-dependent lactate accumulation and secretion into the extracellular microenvironment may lead to acidosis and immunosuppression, favoring immune escape and proliferation [206]. Several other studies consistently showed that glutamine metabolism is critical for tumor growth in several types of cancer, including acute myeloid leukemia [207,208], hepatocellular carcinoma (HCC) [169], pancreatic cancer [173,175], lung cancer [104,157], and neck tumors [185]. Mechanically, the maintenance of a cellular redox state by the synthesis of GSH and ME1-catalyzing NADPH is believed to facilitate cancer cell survival and proliferation [71,209]. Furthermore, a recent study found that glutamine-deriving α-KG mediates the activation of NF-κB signaling through directly binding to IKKs, thereby promoting glucose uptake and tumor growth [163]. As proliferating cells require abundant nutrients, thus shunting their metabolites into anabolic pathways [71], the de novo syntheses of fatty acids and pyrimidines that fuel the duplication of the plasma membrane and nucleic acids, respectively, are also critical in supporting cancer cell growth and proliferation [119,178,185].

### 5.2. Cancer Metastasis

The canonical process of cancer metastasis involves a complex succession of cellular biological events, collectively termed the invasion–metastasis cascade, during which cancer cells in primary tumors invade locally, survive in the vasculature, home into secondary sites, and reinitiate proliferation, thereby generating macroscopic, clinically detectable metastatic colonizations [210]. Redox regulation has long been correlated with cancer metastasis. Several studies showed that endogenous ROS accumulation, induced by antioxidant factor deletion or ROS-producing enzyme activation, mediates the distant metastasis of established tumors, which could be blocked by using exogenous antioxidants NAC or vitamin E [202,211,212]. The wide crosslinks between redox regulation and cancer metastasis continue to be investigated in an increasing number of studies that collectively suggest that the fate decision of cancer cells largely depends on their adaptive changes under oxidative stress. In these contexts, various metabolic factors were found to be involved in the regulation of cancer metastasis, such as antioxidant capacity in a stressful circulating environment, immune modulation, and energy supply for motility.

Several groups have validated the upregulated antioxidant factors within metastatic lesions compared to primary tumors, such as SOD, GCL, G6PD, and NRF2 [211,213,214,215], while separate studies have shown that elevated GSH promotes metastasis in both melanoma and liver cancer [216,217], collectively indicating the importance of increased antioxidant potential during metastasis. Compared to CSC properties, cancer metastasis is more likely to depend on anaerobic glycolysis rather than the PPP (Figure 4D). For example, knockout of TIGAR, a protein supporting PPP activation by hydrolyzing F-2,6BP and F-1,6BP back into F-6P, promoted invasion and metastasis, meanwhile it reduced proliferation and intraepithelial neoplasia lesions in KRAS-driven PDAC [202]. Interestingly, in the same study it was shown that knockout of NRF2 mimicked the effects of TIGAR knockout in promoting invasion and metastasis in KRAS-driven PDAC [202]. Another study showed that NRF2 activation, through KEAP1 loss, also facilitated cell migration and metastasis in KRAS-driven NSCLC [96,168]. In the same study, BACH1-induced metastasis through upregulation of glycolysis can be blocked by treatment with AZD3965, an inhibitor of the lactate secretory channel [168], suggesting the critical role of lactate in mediating cancer metastasis. Lactate secretion has been well documented to promote metastasis through stimulating angiogenesis and immunosuppression [206]. In addition, mtOXPHOS suppression is also believed to facilitate cancer metastasis, potentially because cancer cells that depend on OXPHOS have high endogenous ROS and may be more sensitive to stressful circulating environments [37]. For instance, BACH1-induced metastasis through upregulation of anaerobic glycolysis was partially inhibited by dicholoroacetate, a PDK inhibitor that facilitates pyruvate conversion into acetyl-CoA [168]. Furthermore, direct blocking of OXPHOS through the nonlethal reduction of NAD(+) levels by interfering with nicotinamide phosphoribosyltransferase expression was found to increase the metastasis of breast cancer [218]. Another study found that a switch of metabolic dependency from OXPHOS to the Warburg phenotype by silencing of peroxisome proliferator-activated receptor-gamma coactivator-1α (PGC1α) [34] conferred poorly metastatic melanoma cells with highly invasive potential [219], suggesting that suppression of OXPHOS may promote metastasis at least partially through a feedback potentiation of anaerobic glycolysis and lactate synthesis. In this context, hypoxia is known to play an important role, cancer cells being especially dependent on anaerobic glycolysis to meet essential energetic requirements. Hypoxia-induced HIF activation also leads to fatty acid synthesis and lipid storage, which regulates metastasis by stimulating angiogenesis, immunosuppression, and migration [220,221]. Lipid droplets, in particular, were shown to promote resistance to ROS in hypoxia–reoxygenation [118], which may facilitate the survival of cancer cells that shuttle from hypoxic locations into normoxic circulating environments. In addition, metabolic regulators can also stimulate cancer metastasis in metabolism-independent ways. For example, NF-κB activation was shown to result in the production of VEGF to stimulate angiogenesis and series of chemokines to recruit immune cells into the tumor microenvironment, resulting in the promotion of epithelial-to-mesenchymal transition (EMT), invasion, and metastasis [142].

## 6. Summary and Perspectives

The criteria for “natural selection” apply equally to cancer cells and organisms. Under harmful oxidative stress, cancer cells can survive only if they have sufficient antioxidant capacity or escape from the stressful environment and eventually maintain a redox homeostasis that ensures normal biochemical and cellular biology. As discussed above, whereas abnormally high ROS levels will promote cell death via apoptosis, necroptosis, and/or ferroptosis, the fate decision of cancer cells under oxidative stress is crucially determined by the outcomes of ROS scavenging and adaptive metabolic programs. The heightened metabolic adaptation of cancer cells is closely associated with resistance to chemo- and radiotherapy, as several commonly used anticancer drugs exhibit cytotoxicity at least partially through excessive oxidative stress [47,222]. Notably, electrophiles, the molecules generated simultaneously with ROS, are also responsible for resistance to ROS-producing therapeutic chemicals in cancer therapy. During cellular redox reactions, ROS and reactive electrophilic species are concurrently produced from the same compounds and then attack various proteins in an independent manner [223]. It has been speculated that these electrophilic derivatives may compete with ROS for binding to nucleophilic amino acids, leading to tumor progression and masking the effect of ROS-inducing drugs [223]. Therefore, identifying the protein targets of electrophiles may be a feasible strategy to deal with chemoresistance caused by electrophilic stress. In a recent study, researchers created a library of cysteines reactive to 285 electrophiles through streamlined cysteine activity-based protein profiling (SLC-ABPP) [224]. This may further facilitate electrophilic stress-based therapeutic regimens.

Interestingly, a group of metabolic modulators that also act as ROS sensors enhance the expression of genes for ROS elimination and metabolic phenotype transition, thereby helping cancer cells to adapt to and survive through oxidative stress or to escape from the stressful environment by initiating the metastasis cascade. The dual functions of these ROS sensors and metabolic modulators are supposed to be elaborately controlled, so that cancer cells reach a balance between proliferation and metastasis. Furthermore, these ROS sensors and metabolic modulators can be stimulated by ROS independently of their specific inducers [113,124,142], which means that more than one metabolic modulator may be activated in a certain oxidative condition. This highlights the importance of an overview of the systemic control of metabolic adaptation collaboratively by these metabolic modulators during oxidative stress, especially when it comes to a countereffect of two different metabolic modulators, with a typical example being the fatty acid metabolism regulated by NRF2 and HIF1 [119,164]. Nevertheless, the metabolic remodelings controlled by these different metabolic modulators share major effects, such as upregulation of the PPP, glutamine metabolism, and lactate synthesis, and suppression of mtOXPHOS, all of which are required for cancer cell survival. Furthermore, the PPP and glutaminolysis coupled TCA together provide requisite substrates supporting cancer cell growth and proliferation, while the lactate advantage over mtOXPHOS plays a central role in driving metastasis. These shared processes of metabolic remodeling provide clues for therapy designs targeting enzymes performing critical functions, such as GLUT and SLC1A5 ensuring the uptake of glucose and glutamine, respectively; G6PD and TKT limiting the oxidative PPP and the non-oxidative PPP, respectively; PDK blocking aerobic glycolysis; and GDH controlling glutaminolysis. However, there are still several problems that remain confusing. For example, well-compensated TCA produces large amounts of NADH, leading to its advantage over NAD^+^ due to mtOXPHOS suppression. This indicates that there may be another means of consuming NADH that may be required by cancer cells for survival and proliferation. In this context, NOXs may be a candidate, which were reported to oxidize both NADH and NADPH [225], and to support glycolysis in cancer cells with mitochondrial dysfunction [226]. Furthermore, glutaminolysis is based on double deamination of glutamine, which produces large amounts of ammonia that need to be transported outside the cell. This suggests another important role for pyruvate, which receives an ammonium to be converted into lactamine via the catalyzation of glutamic-pyruvic transaminase (GPT), which is another essential pathway in cancer cells. Notably, cancer cells undergo progression through stepwise transitions, from quiescent tumor initial cells (TIC) to proliferative phenotypes, thus facilitating tumorigenesis, and, in turn, from proliferative phenotypes to quiescent metastatic cells [227]. This may highlight the importance of the high plasticity of metabolic programs in promoting metastasis in cancer; however, this is a subject that needs further investigation.

## Figures and Tables

**Figure 1 antioxidants-11-01324-f001:**
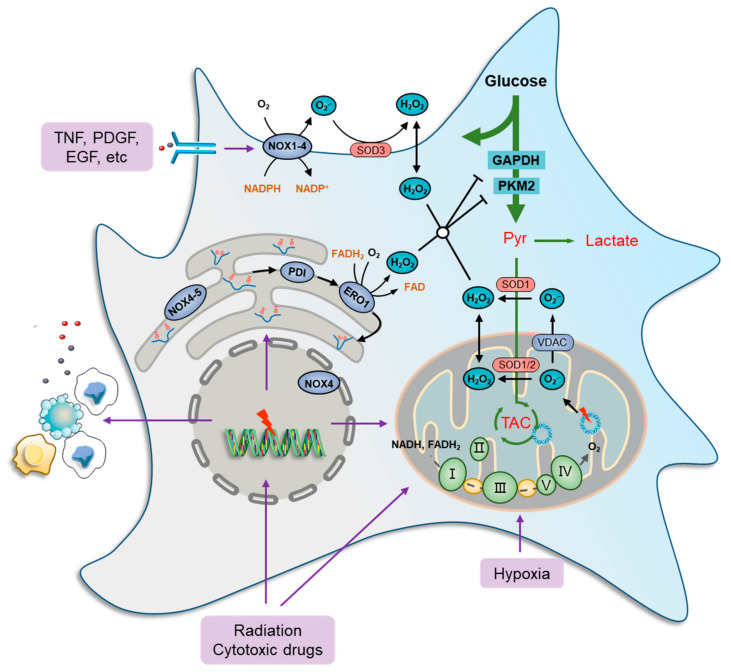
Oxidative stress and metabolic impairment in cancer cells. Mitochondria contribute the most to both physiology- and pathology-derived ROS, of which a fraction is generated by tricarboxylic acid cycle (TCA) enzymes, while the major portion is produced along the electron transport chain (ETC) due to electron leakage at complexes I, II, and III. NOXs are another major source of ROS production in cancer cells. These ROS-generating enzymes function by transmitting one electron from cytosolic NADPH to O_2_ to produce a superoxide anion radical (O_2_^•−^) that can be transformed to H_2_O_2_ immediately by superoxide dismutase (SOD) families. There are seven human NOX homologues, NOX1–5, dual oxidase 1 (DUOX1), and DUOX2, distributed at the cell membrane, mitochondria, ER, and nucleus. They can be activated by a wide variety of ligands, such as tumor necrosis factor (TNF), angiotensin II, platelet-derived growth factor (PDGF), and pro-epidermal growth factor (EGF). The endoplasmic reticulum (ER) serves as a repository wherein nascent proteins are folded and modified, in which disulfide bond formation is essential for the primary structure of proteins and is catalyzed by protein disulfide isomerase (PDI), which can be reduced by ER oxidoreductases, represented by oxidoreductase 1 (ERO1), to generate H_2_O_2_ as a byproduct. Furthermore, several glycolytic enzymes are vulnerable to elevated ROS levels and can be inactivated by redox modification, such as glyceraldehyde-3-phosphate dehydrogenase (GAPDH) and pyruvate kinase M2 isoform (PKM2), which collaboratively cause metabolic stress characterized by a deficiency of carbon sources for ATP generation, lactate secretion, and TCA-driven anabolic biosynthesis.

**Figure 2 antioxidants-11-01324-f002:**
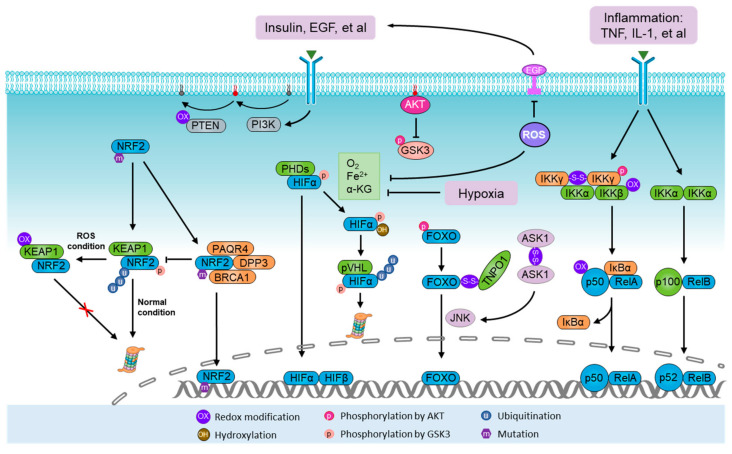
Activation of ROS sensors and metabolic modulators. NRF2-Kelch-like ECH-associated protein 1 (KEAP1) system represents the paradigmatic reactive oxygen species (ROS) sensor apparatus. Under normal conditions, NRF2 is degraded by the KEAP1–Cullin 3 (CUL3) E3 ligase complex. During oxidative stress, the KEAP1 system is directly modified by ROS, which leads to conformational changes in the KEAP1–CUL3 complex, thus disabling ubiquitination and degradation of NRF2. During normoxia, the proline residues of hypoxia-inducible factor α (HIFα) are hydroxylated by prolyl hydroxylases (PHDs) in the presence of oxygen (O_2_), iron (Fe^2+^) and α-ketoglutarate. Under hydroxylation, HIFα is polyubiquitylated by the von Hippel–Lindau tumor suppressor protein (pVHL) complex and guided for proteasomal degradation. Homodimers of IKKγ can be formed through ROS-induced disulfide bonds, thus potentiating its downstream effect on NF-κB. The PI3K-AKT pathway can be stimulated by ROS through phosphorylation and inhibition of tensin homolog (PTEN). In addition, AKT also controls the activity of NRF2, HIFα, NF-κB, and FOXO. The FOXO family are activated by ROS signaling but inhibited by the canonical insulin receptor through PI3K/AKT in the presence of growth factors. Furthermore, ASK1/JNK pathways mediate FOXO activation upon oxidative stress, in which ROS-induced homodimers of ASK1 directly lead to the activation of JNK, a kinase predominantly facilitating the nuclear translocation and activation of FOXO through phosphorylation.

**Figure 3 antioxidants-11-01324-f003:**
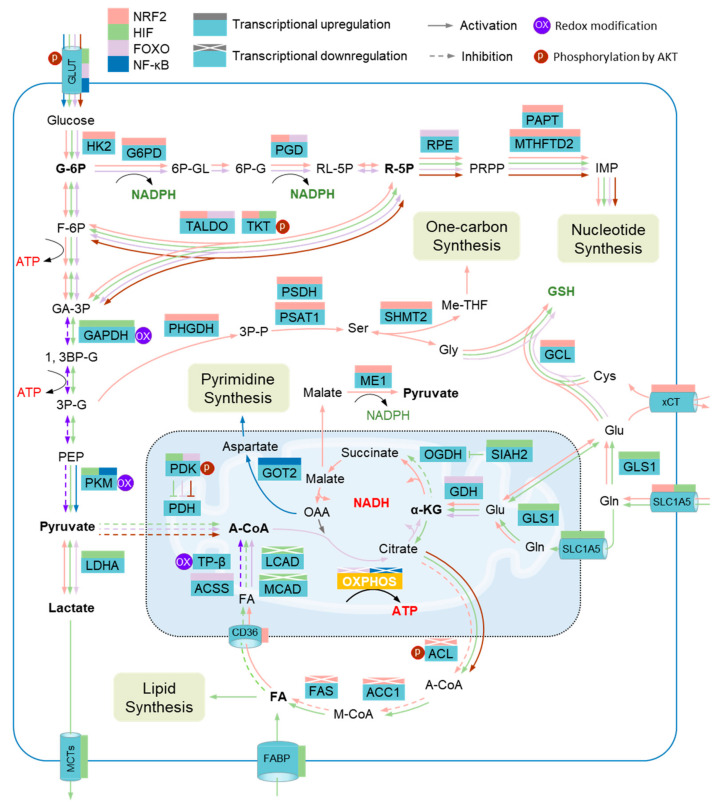
Metabolic adaptation to oxidative stress and metabolic impairment. Several metabolic enzymes are vulnerable to elevated reactive oxygen species (ROS) levels, including glyceraldehyde-3-phosphate dehydrogenase (GAPDH), pyruvate kinase M2 isoform (PKM2), and mitochondrial trifunc-tional protein subunit β (TPβ). Metabolic adaptation during oxidative stress depends on the collaboration of several metabolic modulators that serve to promote metabolic reprograming through regulating the expression of metabolic enzymes. The transcriptions of a host of enzymes in the pentose phosphate pathway (PPP), the serine (Ser)-one carbon pathway, the glutathione pathway (GSH), and glutaminolysis are driven by nuclear factor erythroid 2-related factor 2 (NRF2), including hexokinase 2 (HK2), glucose-6-phosphate dehydrogenase (G6PD), phosphogluconate dehydrogenase (PGD), transaldolase (TALDO), transketolase (TKT), putrescine aminopropyltransferase (PAPT), methylenetetrahydrofolate dehydrogenase 2 (MTHFD2), phosphoglycerate dehydrogenase (PHGDH), phosphoserine phosphatase (PSDH), phosphoserine aminotransferase 1 (PSAT1), serine hydroxymethyltransferase 2 (SHMT2), glutamate-cysteine ligase (GCL), cystine/glutamate transporter (xCT), solute carrier family 1 member 5 (GLC1A5), and malic enzyme 1 (ME1), while several other enzymes involved in the de novo synthesis of fatty acids are transcriptionally inhibited by NRF2, such as acetyl-CoA carboxylase 1 (ACC1), ATP-citrate lyase (ACL), and fatty acid synthase (FAS). Hypoxia-inducible factor (HIF) members promote the expression of enzymes involved in anaerobic glycolysis, de novo synthesis of fatty acids, and glutaminolysis, including glucose transporters (GLUTs), GAPDH, PKM, siah E3 ubiquitin protein ligase 2 (SIAH2), pyruvate dehydrogenase kinase (PDK), lactate dehydrogenase A (LDHA), and glutaminase GLS; on the other hand, HIFs inhibit the expression of enzymes governing the degradation of fatty acids, such as acyl-CoA dehydrogenase long chain (LCAD) and acyl-CoA dehydrogenase medium chain (MCAD). Several enzymes involved in the PPP, degradation of fatty acids, and glutaminolysis, including GLUT, PGD, TALDO, ribulose-5-phosphate-3-epimerase (RPE), acetyl-CoA synthetase (ACSS), and glutamate dehydrogenase (GDH), are transcriptionally activated by forkhead box protein O (FOXO). Nuclear factor-κB (NF-κB) promotes the transcription of PKM and glutamic-oxaloacetic transaminase 2 (GOT2). In addition, the activity of TKT, PDK, and ACL is regulated by Serine/Threonine Kinase 1 (AKT). G-6P, glucose-6-phosphate; 6P-GL, glucono-1,5-lactone-6-phosphate; 6P-G, gluconate-6-phosphate; RL-5P, ribulose-5-phosphate; R-5P, ribose-5-phosphate; IMP, inosine 5’-monophosphate; F-6P, fructose-6-phosphate; GA-3P, glyceraldehyde-3-phosphate; 3P-P, 3-Phosphonooxypyruvate; Gly, glycine; Me-THF, methylenetetrahydrofolate; Cys, cysteine; Glu, glutamate; 1, 3BP-G, glycerate-1, 3-biphosphate; 3P-G, glycerate-3-phosphate; PEP, Phosphoenolpyruvate; MCTs, monocarboxylate transporters; PDH, pyruvate dehydrogenase; A-CoA, acetyl-CoA; OAA, oxaloacetate; α-KG, α-ketoglutarate; OGDH, oxoglutarate dehydrogenase; Gln, glutamine; FA, fatty acid; OXPHOS, oxidative phosphorylation; CD36, fatty acid translocase; M-CoA, malonyl-CoA; FABP, fatty acid binding protein.

**Figure 4 antioxidants-11-01324-f004:**
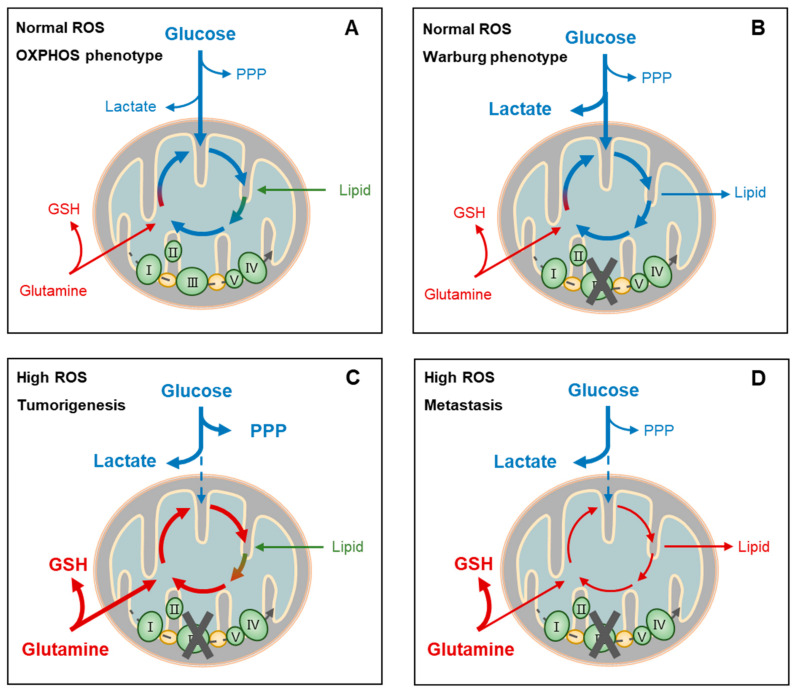
Metabolic adaptation-linked cancer stemness and progression. Cancer cells may present distinct metabolic phenotypes (oxidative phosphorylation, OXPHOS, or Warburg) or biofunctions (tumorigenesis or metastasis) at certain reactive oxygen species (ROS) levels, depending on the circumstances and intrinsic responses. For example, at normal ROS levels, cancer cells of the OXPHOS phenotype depend on aerobic glycolysis, glutaminolysis, and lipolysis for fueling tricarboxylic acid cycle (TCA)-coupled OXPHOS. This highly efficient usage of nutrients for ATP generation facilitates the survival, rather than the proliferation, of cancer cells when facing a shortage of these nutrients (**A**). On the other hand, cancer cells of the Warburg phenotype at normal ROS levels also depend on glycolysis primarily for supporting lactate production and TCA-coupled biosynthesis, rather than OXPHOS, such as the de novo synthesis of amino acids and fatty acids, which is known to facilitate proliferation of cancer cells (**B**). Cancer cells during metastasis and tumor initiation share similarly high ROS conditions and metabolic phenotypes. Extensive ROS blocks aerobic glycolysis, resulting in lactate accumulation (**C**,**D**) and perhaps pentose phosphate pathway (PPP) activation when proliferation is required in tumorigenesis (**C**). On the other hand, glutamine uptake and glutathione (GSH) generation are upregulated (**C**,**D**). In addition, TCA function is recovered by glutaminolysis and lipolysis when proliferation requires reactivation for tumorigenesis (**C**). PPP, pentose phosphate pathway; GSH, glutathione.

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
