# Peer review of "Metabolic Adaptation-Mediated Cancer Survival and Progression in Oxidative Stress"

_antioxidants, 2022, doi:10.3390/antiox11071324_

Round 1

Reviewer 1 Report

It would be great if authors could summarize this review in the end and propose/suggest what is important for future research. 

Reviewer 2 Report

The manuscript “Metabolic Adaptation-mediated Cancer Survival and Progression in Oxidative Stress” provides an overview of metabolic and signalling pathways leading to cancer survival of oxidative stress.

The manuscript is well written and provides a nice overview of complex events triggered by oxidative stress which contribute to cancer survival and metastasis. The authors connect all the intertwining pathways nicely, creating a nice review of events leading to cancer survival and progression. These are qute complex pathways and there are some things that need to be clarified to create a clear picture of these events.

These are some of the issues that need to be clarified:

“…KEAP1 is oxidized and dissociated from NRF2 thus promoting NRF2 stabilization”

KEAP1 is not dissociated from NRF2, but rather, conformational changes occur that disable ubiquitination and newly synthesized NRF2 is free to translocate to nuclei.

Figure 2 is not correct. The position of NRF2 is not clear, all proteins are in cytoplasm, crossing the nuclei membrane, but NRF2 seems to be in the nucleus, which is not the case. In addition, text referring to NRF2 mutations indicates Figure 2, but these events are not really shown in the Figure (for e.g. “Besides, some malignant markers, such as DPP3, PAQR4 and BRCA1, were found to competitively bind to KEAP1 or NRF2, thus interfering their interaction (Figure 2)”).

…noteworthy that AKT also controls other metabolic modulators, including NRF2, HIFα, NF-κB and FOXO as discussed above [17] (Figure 2). This is not discussed above, there is no mention of AKT regulation of NRF2, HIF1alpha or NFkappaB.

Figure 3 needs a clear explanation, the coding should be explained better, are the colour of arrows implying the involvement of specific proteins in their activation or synthesis?

Authors often use the word transactivate. What do they mean by this? For e.g. “glutamyl-cysteinyl ligase (GCL), the rate-limiting enzyme catalyzing GSH synthesis, is also transactivated by NRF2”, NRF2 is the transcription factor for this enzyme, so it activates its transcription, without any intermediate protein transcription/activation (which is implied by the use of “transactivation”).

Reviewer 3 Report

This reviewer considers that this ms represents indeed a significant contribution to the understanding of the oxidative stress signaling in  cancer cells. However,  an adequate level of editing work maybe required to help the readers better understand the key ideas that the authors want to convey.  In addition, the organization of the text may be improved. 

Some minor points to consider:

(1) The word "endogenic"  in line 15 - is likely a misused word. Maybe "endogenous"  ?

 (2) In Introduction, the authors stated in line 32 "......, but cancer cells frequently undergo oxidative stress as a result of .... ".  In fact, oxidative stress is a pathological condition happened to cells after certain events.  So logically, it maybe easier to understand if one states ".... oxidative stress occurs in cancer cells as a result of ....;  alternatively one may also write " cancer cells undergo oxidative stress response triggered by .....". 

(3) In line 68 "Oxidative stress in cancer cells", the authors talk about mitochondria, NOX, and ER. As these 3 terms are not of the same categories, it may be better to reorganized the text based either on the function of the enzymes of interests or the  organelles involved (as NOX exists in many different compartments of cells).

(4) Incomplete legend descriptions: for example in Figure 3, the differences in color codes of the enzymes and the metabolites ? Any differences between the green metabolites vs. red metabolites or black metabolites ? More detailed descriptions of the figures are also recommended to help the readers to grasp the major points of this review. The abbreviations may be placed throughout the text instead of all in the legends. 

Reviewer 4 Report

Reactive oxygen species are known to be tumor promoting agents and many chemotherapeutics generate ROS in oxidative stress. The number of publications in this area is large, and it seems to me that the authors have chosen a reasonable number of important and attractive articles. The review is accompanied by high-quality pictures that help to understand the problem in a comparative and global aspect. I appreciate the value of the review for a better understanding of cancer chemotherapy, since there are still no significant drugs to prevent the destructive action of ROS in the development of tumors. Therefore, the review is of value to specialists in the field of chemotherapy of tumors.

However, the review does not provide a complete explanation of the chemical/biological mechanisms leading to the diversity of the action of ROS during the progression of oxidative stress, namely the onset of electrophilic stress (data published in 2018 and later). Electrophilic molecules generate simultaneously with ROS during chemical transformations of the same compounds in cells. The appearance within minutes of electrophilic derivatives quickly leads to their binding to many proteins, the activation of which can explain the unexpected behavior of cancer cells. The resulting electrophilic derivatives compete with ROS for binding to nucleophilic amino acids, and their detrimental effects may be an important cause of tumor progression. Thus, this event explains the "non-therapeutic" behavior of "ROS-producing therapeutic chemicals" in cancer progression. 

I think, the role of electrophiles should be considered in the context of the review.

Small remarks:

1. Give the names of ROS - lines 107, 154 and correct the chemical designation.

2. What does the CSS abbreviation mean - line 538.

3. It is unclear what normal ROS and high ROS mean in Figure 4. Quantify these terms in the legend.

4. In the chapter "Glutamine dependency" note that this amino acid is sensitive to 37°C. Therefore, this dependence cannot be the result of glutaminolysis in some of the cited studies.

5. Clarify that hydrogen peroxide is an activator of several signaling pathways.

Round 2

Reviewer 1 Report

excellent review

Author Response

Thank you very much for your approval.

Reviewer 2 Report

The authors did nicely modify according to suggestions, but Figure 2 still has a mistake regarding the NRF2. Namely, the NRF2 does not diffuse from KEAP1, when KEAP is oxidatively modified, but rather ubiquitination is disabled but NRF2 is still bi+ound to KEAP1. This enables newly synthesised NRF2 to translocate to the nucleus. This is described in the text, but not in the figure.

Author Response

We are grateful for your precise and valuable suggestions that pretty improved our manuscript. As you suggested, the figure 2 was carefully modified  to show that NRF2 is still binding to KEAP1 when KEAP1 is oxidatively modified, which disables ubiquitination of NRF2, allowing nascent NRF2 to freely translocate into nuclei, as described in the figure legend. We hope this has addressed your concern, and please feel free to point out if there is any other problem. 

Reviewer 4 Report

I think that the authors answered the questions posed, including showing to some extent the possible role of electrophilic molecules in the progression of cancer under oxidative stress.

Therefore, I recommend the manuscript for publication.

Author Response

Thank you very much for your approval.